# Glucose control and outcomes in diabetic and nondiabetic patients treated with targeted temperature management after cardiac arrest

Hyo Jin Bang[1], Chun Song Youn[1]*, Kyu Nam Park[1], Sang Hoon Oh[1], Hyo Joon Kim[1], Soo Hyun Kim[2], Sang Hyun Park[3]

1 Department of Emergency Medicine, Seoul St. Mary Hospital, College of Medicine, The Catholic University of Korea, Seoul, Republic of Korea, 2 Department of Emergency Medicine, Eunpyeong St. Mary Hospital, College of Medicine, The Catholic University of Korea, Seoul, Republic of Korea, 3 Department of Emergency Medicine, Yeouido St. Mary Hospital, College of Medicine, The Catholic University of Korea, Seoul, Republic of Korea

* ycs1005@catholic.ac.kr

**Data Availability Statement:** All relevant data are within the manuscript and its Supporting Information files.

## Abstract

Hyperglycemia is commonly observed in critically ill patients and postcardiac arrest patients, with higher glucose levels and variability associated with poorer outcomes. In this study, we aim to compare glucose control in diabetic and nondiabetic patients using glycated hemoglobin (HbA1c) levels, providing insights for better glucose management strategies. This retrospective observational study was conducted at Seoul St. Mary's Hospital from February 2009 to May 2022. Blood glucose levels were measured hourly for 48 h after return of spontaneous circulation (ROSC), and a glucose management protocol was followed to maintain arterial blood glucose levels between 140 and 180 mg/dL using short-acting insulin infusion. Patients were categorized into four groups based on diabetes status and glycemic control. The primary outcomes assessed were neurological outcome and mortality at 6 months after cardiac arrest. Among the 332 included patients, 83 (25.0%) had a previous diabetes diagnosis, and 114 (34.3%) had an HbA1c of 6.0% or higher. At least one hyperglycemic episode was observed in 314 patients (94.6%) and hypoglycemia was found in 63 patients (19.0%) during 48 h. After the categorization, unrecognized diabetes was noticed in 51 patients with median HbA1c of 6.3% (interquartile range [IQR] 6.1–6.6). Patients with inadequate diabetes control had the highest initial HbA1c level (7.0%, IQR 6.5–7.8) and admission glucose (314 mg/dL, IQR 257–424). Median time to target glucose in controlled diabetes was significantly shorter with the slowest glucose reducing rate. The total insulin dose required to reach the target glucose level and cumulative insulin requirement during 48 h were different among the categories (p <0.001). Poor neurological outcomes and mortality were more frequently observed in patients with diagnosed diabetes. Occurrence of a hypoglycemic episode during the 48 h after ROSC was independently associated with poor neurologic outcomes (odds ratio [OR] 3.505; 95% confidence interval [CI], 2.382–9.663). Surviving patients following cardiac arrest exhibited variations in glucose hemodynamics and outcomes according to the categories based on their preexisting diabetes status and

**Funding:** The authors received no specific funding for this work.

**Competing interests:** The authors have declared that no competing interests exist.

glycemic condition. Specifically, even experiencing a single episode of hypoglycemia during the acute phase could have an influence on unfavorable neurological outcomes. While the classification did not directly affect neurological outcomes, the present results indicate the need for a customized approach to glucose control based on these categories.

## Introduction

Efforts to improve survival and neurological outcomes in patients who achieved return of spontaneous circulation (ROSC) after cardiac arrest (CA) have been made through various studies [1–3]. Targeted temperature management (TTM) is applied to unconscious patients after CA to attain a neuroprotective effect in accordance with international guidelines [3, 4]. Although much evidence is emerging to optimize general intensive care management, such as blood pressure targets, gas-exchange parameters, and ventilator settings, the specific target is still uncertain [5–8]. In the same manner, ideal glucose management is not completely known. In the absence of an optimal target range of glucose during postcardiac arrest care, a range of 140–180 mg/dL is recommended along with avoidance of hypoglycemia, and the same approach is taken for the critically ill [3, 4, 9].

Hyperglycemia is commonly observed in critically ill patients and can have an impact on their outcomes [10–14]. Prolonged exposure to hyperglycemia in diabetic patients has been shown to cause microvascular and macrovascular complications [15–17]. Recently, it has been identified as a risk factor for a different set of complications and can have an impact on patient morbidity and mortality [18]. The fact that hyperglycemia can impact outcomes has also been observed in postcardiac arrest [19–24]. Through these studies, it has been confirmed that higher glucose levels immediately after ROSC are associated with poor outcomes; there is also a correlation between glucose variability and poor outcomes.

The relationship between diabetes and poor neurologic outcome was already demonstrated in a meta-analysis with OHCA survivors [25]. As personalized glucose control strategies based on preadmission glycemic control have become increasingly important in the critically ill, it is also important to control glucose levels based on a patient's glycemic status during postcardiac arrest care [26]. Our hypothesis was there would be differences in outcomes and variables according to the glycemic status of patients and whether they were diagnosed with diabetes. We wondered if there were any differences in outcomes and variables according to the glycemic status of patients and whether they were diagnosed with diabetes. Our final aim was to compare the glucose hemodynamics of these subgroups and, if differences exist, provide a basis for more ideal glucose control for each group.

## Material and methods

### Study design and participants

This retrospective observational study was conducted in a single center, the Seoul St. Mary's Hospital, which is a regional emergency medical center of a tertiary hospital, from February 2009 to May 2022. Comatose patients who had ROSC after CA were treated with TTM. All adults ($\geq$ 18 years) resuscitated after out-of-hospital cardiac arrest (OHCA) and maintained in ROSC for longer than 20 min were included for analysis. Patients with active intracranial bleeding, acute stroke, known limitations in therapy and a do-not-attempt resuscitation order, known prearrest CPC score of 3 or 4 and body temperature of 30˚C on admission were not

treated with TTM. Patients with missing data for blood HbA1c levels at admission and neuro-logic outcomes at 6 months after ROSC were excluded. Implementation of TTM, including the target temperature setting, TTM duration, and TTM methods, was in accordance with a preestablished protocol [27].

The study was approved by the Institutional Review Boards (IRBs) (KC22RASI0954) of the Seoul St. Mary's Hospital. Informed consent was waived due to the retrospective nature of the study. The data was accessible on December 23, 2022 for research purposes. While only HJ Kim accessed and anonymized the data before anonymization, the other authors could not access to information that could identify individual participants.

## Glucose management protocol

The glucose level was measured from blood samples obtained from an arterial catheter using a handheld glucose measurement device every hour until 48 h after ROSC. The results were automatically stored in the clinical information system. Avoidance of glucose-containing solu-tions was recommended unless hypoglycemia was present. Hypoglycemia was defined as a blood glucose value less than 72 mg/dL [28]. Short-acting insulin infusion was used to main-tain the arterial blood glucose level at 140 to 180 mg/dL. The same protocol was applied for all patients, and the details of the protocol were described in a previous study [29]. In this study, we classified patients into two groups: those who had been diagnosed with diabetes and those who had not. We then further classified patients in these two groups into subgroups based on their preexisting levels of glucose control using glycated hemoglobin (HbA1c). We used HbA1c to assess preexisting levels of glucose control [30–32]. Clinical guidelines generally advocate for an HbA1c threshold of ≥6.5% for diagnosing diabetes and a range of 5.7 to 6.4% for identifying prediabetes. However, Silverman et al. have suggested that in acute-care settings such as emergency departments, an HbA1c of 5.7% is the optimal screening cutoff for predia-betes, while 6% is optimal for diagnosing diabetes [33, 34]. Consequently, we established a cut-off value of HbA1c at 6.0% to assess the adequacy of glucose control and diagnose diabetes.

## Data collection

The participants' medical records were reviewed according to the Utstein Style Criteria for reporting OHCA [35]. We extracted the following baseline clinical data: sex, age, comorbidi-ties (acute myocardial infarction (AMI), angina pectoris, congestive heart failure (CHF), hypertension, diabetes mellitus (DM) and renal disease), cause of arrest, initial CA rhythm (shockable or nonshockable), presence of witness, bystander cardiopulmonary resuscitation (CPR) and total anoxic time (time from collapse to ROSC). The glucose-related variables were extracted as follows: HbA1c at admission, initial glucose level, glucose variability within 48 h after ROSC insulin dose to target glucose level (<180 mg/dL) within 48 h after ROSC and time to reach target glucose level. The initial HbA1c was measured immediately after ROSC. The glucose variability included the median, range, and mean value of the glucose. The range was calculated as the difference between the maximum and minimum blood glucose values during 48 h [20]. The time to reach the target glucose level was defined as the first time the target range was reached. The total insulin dose meant cumulative insulin to reach the target glucose at the first time. The glucose reduction rate was calculated as the ratio of the difference between the initial glucose level and the first value in the range of target glucose levels to the time taken to reach the target level.

In this study, we categorized all included patients into four groups as follows: inadequately controlled diabetes, controlled diabetes, unrecognized diabetes and no diabetes. We set an opti-mal HbA1c cutoff of 6.0% as a diagnostic measure for diabetes and the average glycemic status

during the previous 1 to 3 months [36]. With the cutoff, the inadequately controlled diabetes group was defined as patients diagnosed with diabetes before CA but with an initial HbA1c of 6.0% or higher. The controlled diabetes was defined as patients with diabetes before CA with an initial HbA1c less than 6.0%. The unrecognized diabetes patients were never diagnosed with diabetes but had an initial HbA1c of 6.0% or higher. Finally, no diabetes was defined as patients were never diagnosed with diabetes but had an initial HbA1c less than 6.0%.

## Outcome measures

The primary outcomes were neurological outcome defined by a Cerebral Performance Category (CPC) and death at 6 months after CA. The CPC scale spans from 1 to 5: 1 signifies good cerebral performance or slight disability, 2 indicates moderate disability with independence in daily activities, 3 denotes severe disability requiring assistance from others, 4 signifies a coma or vegetative state, and 5 signifies death or brain death. A good neurological outcome was defined as CPC 1–2, while a poor neurological outcome was defined as a CPC score of 3–5. The researcher contacted surviving discharged patients or their relatives for follow-up. Face-to-face visits or telephone interviews were recommended.

## Statistical analysis

All data are displayed as numbers and percentages for categorical variables and as medians with interquartile ranges (IQRs) for continuous variables. Comparisons of categorical variables between the groups were made using the Chi-square test or Fisher's exact test. After being tested for normal distribution, continuous variables were compared using Student's t test or Wilcoxon's rank-sum tests. To assess independent predictors of poor neurologic outcome and death, we included all variables with p-values less than 0.05 in univariate analyses, along with a set of potential confounders derived from published studies, in a multivariate logistic regression analysis.. The considered confounders included witness status, bystander CPR, and the initial presence of a shockable rhythm [3, 37, 38]. The odds ratios (ORs) with 95% confidence intervals (CIs) were calculated. Statistical analyses were performed using SPSS version 24.0 (SPSS, Chicago, IL, USA). A p value <0.05 was considered statistically significant.

## Results

### Enrollment and characteristics of patients

During the study period, 407 patients were admitted after OHCA and treated with TTM. Seventy-five were excluded due to missing data for initial HbA1c and 6-month outcome. The remaining 332 patients were categorized according to their diabetes diagnosis and HbA1c. Eighty-three patients (25.0%) were previously diagnosed with diabetes, and 114 patients (34.3%) had an HbA1c of 6.0% or higher. Accordingly, 63 patients with inadequately controlled diabetes, 20 patients with controlled diabetes, 51 patients with unrecognized diabetes and 198 patients with no diabetes were included in the groups (Fig 1).

Among the 332 enrolled patients, 72.9% were male, the median age was 56.5 years old, 41.6% had a noncardiac cause of arrest, 62.0% had a nonshockable initial rhythm, 32.8% had unwitnessed arrest, 63.3% received bystander CPR, and the median anoxic time was 36.7 min. Poor neurological outcome and no survival at 6 months after CA occurred in 232 (69.9%) patients and 197 (59.3%) patients, respectively.

Comparing the categorization, patients classified as inadequately controlled diabetes, controlled diabetes, unrecognized diabetes and no diabetes were significantly older in order. Acute myocardial infarction, hypertension, and renal disease were significantly more common

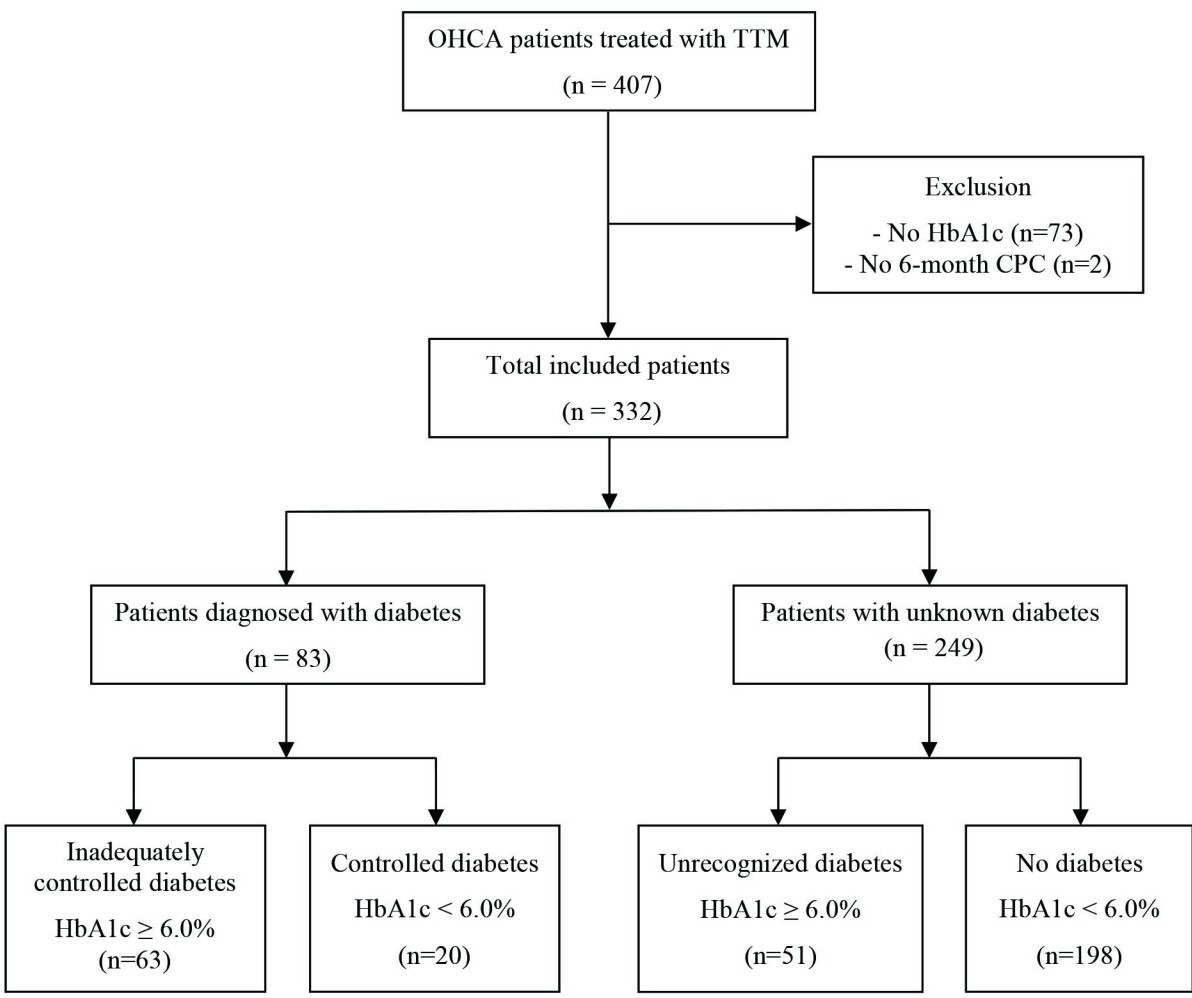

**Fig 1. Flow chart of the study.**

in patients with known diabetes. While nonshockable initial rhythm was more frequently observed in inadequately controlled diabetes (77.8%) and controlled diabetes (80.0%), there were no significant differences in other variables related to CA (Table 1).

## Glucose hemodynamics and outcomes

Comparisons of variables related to glucose are described in Table 2. Admission glucose, mean and median values of glucose level were significantly higher in the groups of inadequately controlled diabetes and unrecognized diabetes. Range of initial glucose level was higher in known diabetes patients. In particular, over 90% of patients with poor glucose control (HbA1c ≥6.0%) showed hyperglycemia at admission. Most patients (94.6%) experienced at least one hyperglycemic episode over 48 h, while a hypoglycemic episode was observed in one-fifth of all patients. Patients with inadequately controlled diabetes had the longest time to target glucose, the most insulin dose to target glucose and the most cumulative insulin requirement to maintain target glucose during the 48 h after ROSC. The time to reach target glucose according to the categorization is presented in Fig 2. Patients in the groups of inadequately controlled diabetes and unrecognized diabetes took longer to arrive at target glucose, and more doses of

**Table 1. Baseline characteristics according to the categorization.**

| | No diabetes | Inadequately controlled diabetes | Controlled diabetes | Unrecognized diabetes | P value |
|---|---|---|---|---|---|
| | n = 198 | n = 63 | n = 20 | n = 51 | |
| Male, n (%) | 135 (68.2) | 52 (82.5) | 13 (65.0) | 42 (82.4) | 0.043 |
| Age, y | 49 (40.0–63.0) | 69 (58.0–75.0) | 66 (56.3–70.8) | 58 (50.0–70.0) | < 0.001 |
| History of previous cardiac arrest, n (%) | 1 (0.5) | 2 (3.2) | 0 (0) | 2 (3.9) | 0.185 |
| History of AMI, n (%) | 5 (2.5) | 9 (14.3) | 4 (20.0) | 5 (9.8) | 0.001 |
| History of angina pectoris, n (%) | 9 (4.5) | 9 (14.3) | 1 (5.0) | 3 (5.9) | 0.058 |
| History of CHF, n (%) | 4 (2.0) | 2 (3.2) | 0 (0) | 1 (2.0) | 0.852 |
| History of hypertension, n (%) | 40 (20.2) | 40 (63.5) | 13 (65.0) | 22 (43.1) | < 0.001 |
| History of renal disease, n (%) | 7 (3.5) | 13 (20.6) | 9 (45.0) | 1 (2.0) | < 0.001 |
| Noncardiac cause arrest, n (%) | 87 (43.9) | 25 (39.7) | 11 (55.0) | 15 (29.4) | 0.162 |
| Initial nonshockable rhythm, n (%) | 116 (58.6) | 49 (77.8) | 16 (80.0) | 25 (49.0) | 0.003 |
| Unwitnessed arrest, n (%) | 71 (35.9) | 16 (25.4) | 5 (25.0) | 17 (33.3) | 0.397 |
| No bystander CPR, n (%) | 63 (31.8) | 30 (47.6) | 7 (35.0) | 22 (43.1) | 0.102 |
| Total anoxic time[a], min | 33.5 (20.0–49.3) | 32.0 (16.5–42.0) | 33.5 (18.0–48.0) | 37.0 (23.0–43.0) | 0.648 |
| Poor neurological outcome, n (%) | 132 (66.7) | 52 (82.5) | 18 (90.0) | 30 (58.8) | 0.006 |
| No survival at 6 month, n (%) | 111 (56.1) | 46 (73.0) | 16 (80.0) | 24 (47.1) | 0.006 |

Data are presented as n (%) for categorical variables and as medians (interquartile range, IQR) for continuous variables.

[a]Defined as the time interval between arrest and ROSC

Abbreviations: AMI, acute myocardial infarction; CHF, congestive heart failure; CPR, cardiopulmonary resuscitation

**Table 2. Comparisons of glucose-related variables.**

| | No diabetes | Inadequately controlled diabetes | Controlled diabetes | Unrecognized diabetes | p value |
|---|---|---|---|---|---|
| | n = 198 | n = 63 | n = 20 | n = 51 | |
| Initial HbA1c, % | 5.4 (5.2–5.6) | 7.0 (6.5–7.8) | 5.6 (5.3–5.8) | 6.3 (6.1–6.6) | < 0.001 |
| Admission glucose, mg/dL | 259.0 (200.0–315.0) | 314.0 (257.0–424.0) | 248.5 (166.5–464.2) | 270.0 (195.0–336.0) | < 0.001 |
| Mean glucose during 48 h, mg/dL | 150.9 (136.7–174.5) | 184.4 (160.0–217.2) | 158.0 (138.5–197.0) | 173.9 (143.2–205.3) | < 0.001 |
| Median glucose during 48 h, mg/dL | 138.5 (123.9–159.6) | 160.0 (145.5–194.0) | 135.5 (128.4–168.5) | 156.0 (135.0–190.0) | < 0.001 |
| Range glucose during 48 h, mg/dL | 190.5 (141.8–258.0) | 263.0 (184.0–391.0) | 266.5 (181.0–352.5) | 200.0 (141.0–286.0) | < 0.001 |
| At least one hyperglycemic episode during 48 h, n (%) | 183 (92.4) | 63 (100.0) | 19 (95.0) | 49 (96.1) | 0.131 |
| At least one hypoglycemic episode during 48 h, n (%) | 36 (18.2) | 12 (19.0) | 4 (20.0) | 11 (21.6) | 0.957 |
| Hyperglycemia at admission, n (%) | 159 (80.3) | 58 (92.1) | 13 (65.0) | 47 (92.2) | 0.006 |
| Time to target glucose, h | 5.0 (2.0–8.0) | 8.5 (6.0–15.0) | 3.0 (0.0–11.0) | 7.0 (4.0–10.8) | < 0.001 |
| Glucose reducing rate, mg/h | 22.5 (12.0–37.8) | 18.3 (11.3–23.3) | 13.6 (8.8–44.2) | 14.3 (8.5–28.1) | 0.064 |
| Total insulin dose to target glucose, IU | 0 (0.0–8.0) | 15 (6.0–40.0) | 2.5 (0.0–19.5) | 3 (0.0–19.0) | < 0.001 |
| Cumulative insulin requirement during 48 h, IU | 7.5 (0.0–33.0) | 50 (30.0–80.0) | 30 (8.5–41.8) | 22 (0.0–55.0) | < 0.001 |

Data are presented as n (%) for categorical variables and as medians (interquartile range, IQR) for continuous variables.

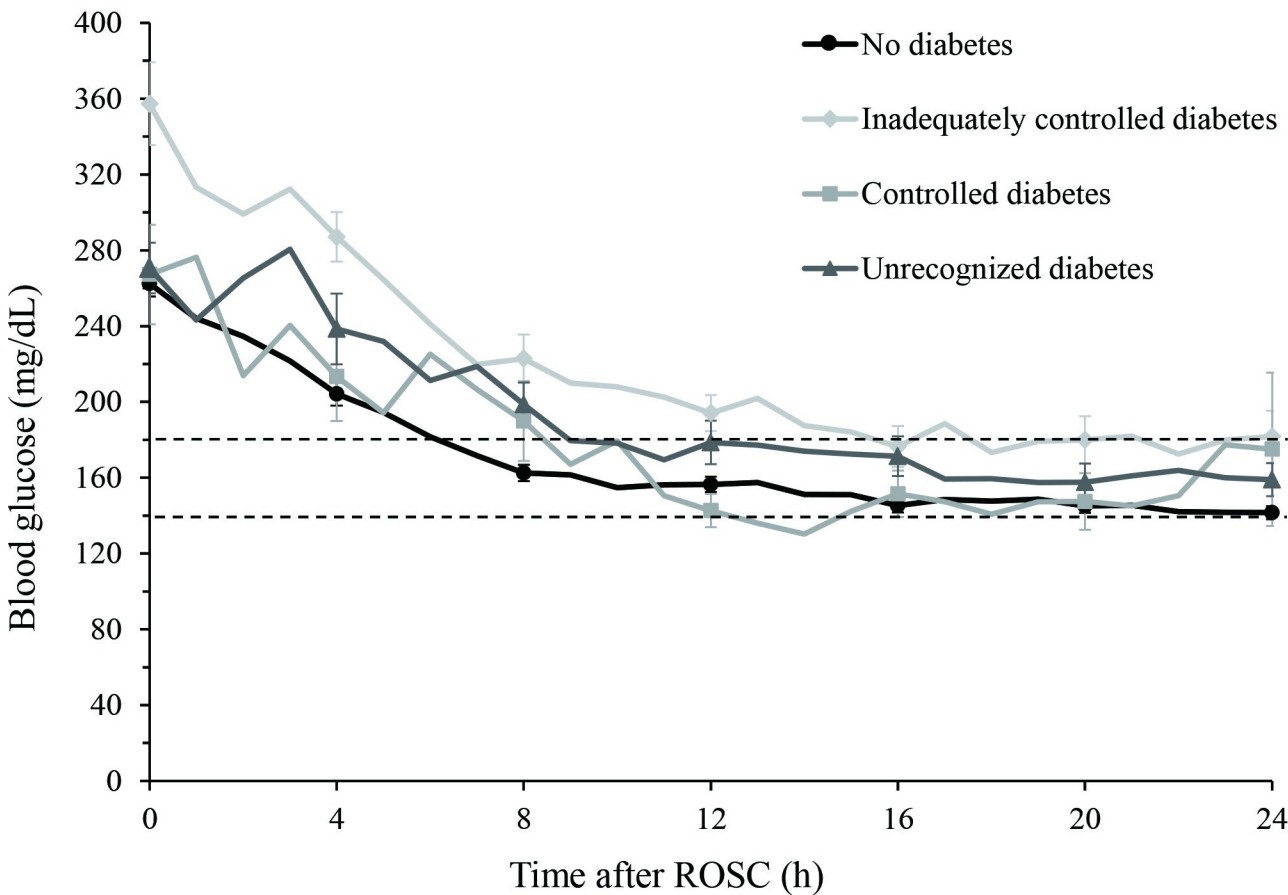

**Fig 2. Time to reach target glucose according to the categories.** Each trend line represents the declining trend in glucose levels for each group, with mean values and standard deviations indicated.

insulin were used than in the groups of controlled diabetes and no diabetes. The time to target glucose, glucose reducing rate and total insulin dose to target glucose were significantly different between the groups. Poor neurologic outcome and death were significantly more frequently observed in patients with diagnosed diabetes.

Table 3 shows the comparison of variables related to glucose level with neurological outcome and survival at 6 months after ROSC. Higher initial glucose levels and the presence of at least one hypoglycemic episode were significantly found in patients with poor neurologic outcomes and death. Additionally, there were significant differences among the groups based on categorization in neurological outcome and death ($p < 0.05$).

## Logistic regression analysis

In univariate logistic regression analysis, older age, higher proportion of diabetes, noncardiac etiology, nonshockable initial rhythm and unwitnessed arrest, longer total anoxic time, higher admission glucose, and presence of hypoglycemic episode were associated with poor neurologic outcome and death (S1 Table). In particular, in terms of death, the presence of comorbidities (hypertension and renal disease), no bystander CPR and higher initial HbA1c were additionally associated. Diagnosed diabetes was significantly associated with poor neurologic outcomes and death, whether well controlled or inadequately controlled.

**Table 3. Characteristics according to the primary outcomes.**

| | Neurological outcomes | | *p-value* | Survival | | *p value* |
|---|---|---|---|---|---|---|
| | **Good (n = 100)** | **Poor (n = 232)** | | **Yes (n = 135)** | **No (n = 197)** | |
| Initial HbA1c, % | 5.6 (5.3–6.1) | 5.7 (5.3–6.4) | 0.189 | 5.6 (5.3–6.1) | 5.7 (5.4–6.4) | 0.053 |
| Admission glucose, mg/dL | 243.5 (191.3–286.5) | 288.0 (215.5–355.0) | < 0.001 | 252.0 (199.0–306.0) | 288.0 (208.5–355.5) | 0.005 |
| Mean glucose during 48 h, mg/dL | 151.9 (138.5–173.5) | 161.3 (145.5–194.6) | 0.004 | 153.1 (138.4–175.6) | 162.3 (146.7–197.8) | 0.001 |
| Median glucose during 48 h, mg/dL | 139.3 (123.9–160.5) | 147.8 (130.0–180.0) | 0.001 | 140.0 (125.5–161.0) | 147.5 (131.0–183.0) | < 0.001 |
| Range glucose during 48 h, mg/dL | 168.5 (132.3–231.5) | 219.0 (165.0–301.0) | 0.019 | 184.0 (134.0–258.0) | 222.0 (165.0–307.0) | 0.059 |
| At least one hyperglycemic episode during 48 h, n (%) | 93 (93.0) | 221 (95.3) | 0.404 | 127 (94.1) | 187 (94.9) | 0.737 |
| At least one hypoglycemic episode during 48 h, n (%) | 9 (9.0) | 54 (23.3) | 0.002 | 15 (11.1) | 48 (24.4) | 0.002 |
| Hyperglycemia at admission, n (%) | 81 (81.0) | 196 (84.5) | 0.434 | 113 (83.7) | 164 (83.2) | 0.913 |
| Time to target glucose, h | 5.0 (2.0–9.0) | 6.0 (3.0–9.0) | 0.146 | 5.0 (3.0–9.0) | 6.0 (2.3–9.8) | 0.288 |
| Glucose reducing rate, mg/h | 16.3 (8.6–30.5) | 20.3 (11.9–36.7) | 0.037 | 18.5 (8.9–35.4) | 20.3 (12.4–35.7) | 0.109 |
| Total insulin dose to target glucose, IU | 0 (0.0–12.8) | 4 (0.0–17.5) | 0.053 | 0 (0.0–14.0) | 4 (0.0–19.0) | 0.080 |
| Cumulative insulin requirement during 48 h, IU | 13.0 (0.0–47.8) | 20.5 (0.25–52.8) | 0.117 | 15.0 (0.0–50.0) | 22 (2.0–53.5) | 0.122 |
| Category | | | 0.010 | | | 0.006 |
| No diabetes, n (%) | 66 (66.0) | 132 (56.9) | | 87 (64.4) | 111 (56.3) | |
| Inadequately controlled diabetes, n (%) | 11 (11.0) | 52 (22.4) | | 17 (12.6) | 46 (23.4) | |
| Controlled diabetes, n (%) | 2 (2.0) | 18 (7.8) | | 4 (3.0) | 16 (8.1) | |
| Unrecognized diabetes, n (%) | 21 (21.0) | 30 (12.9) | | 27 (20.0) | 24 (12.2) | |

Data are presented as n (%) for categorical variables and as medians (interquartile range, IQR) for continuous variables.

Age, noncardiac etiology and nonshockable initial rhythm were associated with poor neurologic outcome and death in multivariate logistic regression (Table 4). Specifically, the nonshockable initial rhythm was highly associated with poor neurologic outcome (OR 5.880, 95% CI, 2.771–12.476) and death (OR 4.725, 95% CI 2.333–9.572). A hypoglycemic episode during the 48 h after ROSC was independently associated with poor neurologic outcomes (OR 3.505; 95% CI, 2.382–9.663). The categorization according to diabetes was not related to the outcomes in multivariate logistic regression analysis.

## Discussion

In this study, we categorized OHCA patients who received TTM based on their diabetes status and glycemic control reflected by HbA1c and compared the differences in glucose hemodynamics among the groups. Indeed, there were differences in initial glucose levels and glucose variability (mean, median, and range) during the first 48 h as well as the time required to reach the target glucose and the insulin dose. Ultimately, there were significant differences in the 6-month outcomes among the groups. Additionally, initial glucose levels and hypoglycemic episode were significantly associated with poor neurological outcomes.

The relationship between a patient's glucose level and outcomes has been studied in previous studies [19–22]. Additionally, HbA1c was analyzed in patients treated with TTM after OHCA [34, 39]. We classified patients previously diagnosed with diabetes into four groups: those with controlled diabetes, inadequately controlled diabetes, unrecognized diabetes, and no diabetes with adequate glucose control. Patients with inadequately controlled diabetes showed the highest levels of HbA1c, admission glucose, mean glucose, and median glucose, while patients in the unrecognized diabetes group followed. HbA1c, which we used to determine glycemic status, is a well-known marker that reflects glucose control over the previous three months [36]. In our study, we were able to identify that more effort was needed when

**Table 4. Multivariate logistic regression analysis.**

| Poor neurologic outcome | OR | 95% CI | p value |
|---|---|---|---|
| Age ≥65 | 2.780 | 1.200–6.445 | 0.017 |
| Noncardiac cause arrest | 5.239 | 2.113–12.989 | <0.001 |
| Initial nonshockable rhythm | 6.439 | 2.950–14.055 | <0.001 |
| Unwitnessed arrest | 0.874 | 0.357–2.137 | 0.768 |
| No bystander CPR | 0.689 | 0.323–1.471 | 0.336 |
| Total anoxic time | 1.063 | 1.038–1.087 | <0.001 |
| Admission glucose | 1.004 | 0.999–1.009 | 0.154 |
| Mean glucose during 48 h | 0.995 | 0.960–1.032 | 0.793 |
| Median glucose during 48 h | 1.014 | 0.983–1.046 | 0.385 |
| Range glucose during 48 h | 1.001 | 0.996–1.006 | 0.699 |
| At least one hypoglycemic episode during 48 h | 3.522 | 1.096–11.322 | 0.035 |
| Categorization | | | 0.573 |
| No diabetes | ref | ref | ref |
| Inadequately controlled diabetes | 1.041 | 0.361–3.005 | 0.940 |
| Controlled diabetes | 2.625 | 0.352–19.568 | 0.346 |
| Unrecognized diabetes | 0.636 | 0.245–1.650 | 0.352 |
| No survival | | | |
| Age ≥65 | 2.242 | 1.061–4.739 | 0.034 |
| History of hypertension | 2.124 | 1.012–4.461 | 0.046 |
| History of renal disease | 1.166 | 0.345–3.941 | 0.805 |
| Noncardiac cause arrest | 2.569 | 1.234–5.351 | 0.012 |
| Initial nonshockable rhythm | 4.838 | 2.371–9.871 | <0.001 |
| Unwitnessed arrest | 1.149 | 0.542–2.436 | 0.718 |
| No bystander CPR | 0.943 | 0.489–1.821 | 0.862 |
| Total anoxic time | 1.059 | 1.039–1.080 | <0.001 |
| Admission glucose | 1.000 | 0.996–1.004 | 0.867 |
| Mean glucose during 48 h | 1.013 | 0.982–1.045 | 0.426 |
| Median glucose during 48 h | 0.999 | 0.973–1.026 | 0.955 |
| Range glucose during 48 h | 0.999 | 0.994–1.004 | 0.683 |
| At least one hypoglycemic episode during 48 h | 2.509 | 0.959–6.565 | 0.061 |
| Categorization | | | 0.385 |
| No diabetes | ref | ref | ref |
| Inadequately controlled diabetes | 0.829 | 0.315–2.179 | 0.703 |
| Controlled diabetes | 1.522 | 0.363–6.388 | 0.566 |
| Unrecognized diabetes | 0.479 | 0.186–1.229 | 0.126 |

applying insulin therapy during acute treatment in patients with inadequately controlled glycemic status. It can be estimated that patients in the inadequately controlled diabetes and unrecognized diabetes groups were exposed to chronic hyperglycemia. Lowering their glucose levels required more time and larger amounts of insulin. The assumptions to explain this are as follows: 1. Diabetic patients have adapted to chronic hyperglycemia and have resistance to lowering glucose with the same insulin dose due to insulin resistance and beta-cell secretory defects [40, 41]. Especially in patients who have already been diagnosed with and treated for diabetes, despite the intervention of glucose-lowering medication, it might be expected that more effort would be required to reach the target because of their being in a state of poor glycemic control. 2. We could consider the additive effect of stress-induced hyperglycemia. Stress hyperglycemia is a transient change in glucose from baseline during illness regardless of the patient's diabetes

status [10]. Stress hyperglycemia is induced by highly complex mechanisms of counterregulatory hormones such as catecholamine, growth hormone, cortisol, and cytokines, and in patients with poor glycemic control, there can be a synergistic effect, leading to sustained hyperglycemia [10].

It was found that the mean and median values of glucose over 48 h, known as glycemic variability, differ among categories depending on a patient's diabetes status and glycemic control status. Glycemic variability is already known as an important factor for outcomes in critically ill patients [11, 42, 43]. In fact, studies have been conducted on the relationship between glycemic variability and outcomes in CA patients [20, 44]. One study compared the relationship between mean glucose and poor outcomes over observation periods of 36, 48, and 96 h and found that it affected outcomes regardless of the observation period [19]. Ola Borgquist et al. showed that the higher the median glucose, the more likely poor neurologic outcomes would occur [20]. Furthermore, in our study, glucose range also differed among groups and was found to be associated with primary outcomes in univariate analysis. Therefore, stabilizing glucose variability according to a patient's condition may affect the outcomes in addition to their initial glucose level. Our study suggests that different approaches may be necessary based on the patient's glycemic status and presence of diabetes. Despite numerous investigations into conventional versus individualized glucose control for critically ill patients, conclusive evidence remains elusive. [26, 45]. A specific study on diabetic ketoacidosis patients indicated that a combination of intravenous insulin and subcutaneous glargine showed a trend towards faster resolution and shorter hospital stays, though statistically insignificant [46]. This highlights the necessity for more research on insulin types, doses, and target glucose levels tailored to individual characteristics, with a crucial emphasis on preventing hypoglycemia.

We found that there were differences in the time and speed to reach target glucose levels among each group of patients, especially in patients with HbA1c levels of 6.0% or higher, who took longer to reach target glucose levels. However, the time to target glucose, glucose reducing rate, and insulin dose to target dose were not independent predictors of primary outcomes in our study. However, in a study by JH Woo et al., a faster time to reach target glucose levels was associated with favorable outcomes in OCHA patients who received TTM [47]. We speculated that this difference may be because our study did not include factors related to TTM in the multivariate analysis. TTM itself can induce insulin resistance, which can affect glucose hemodynamics, as reported by Sah Pri, Azurahisham, et al. [48].

The occurrence of at least one episode of hypoglycemia during 48 h was associated with a poor neurological outcome after 6 months, both in univariate and multivariate analyses. It is already well known that hypoglycemia is associated with increased mortality in critically ill patients [49, 50]. In addition, a previous observational study analyzing OHCA patients treated with TTM also confirmed that hypoglycemia is related to poor neurological outcome and mortality [23, 51]. This supports the guidelines to prevent hypoglycemia [3, 4]. Moreover, in our study, even though a broader hypoglycemia standard (72 mg/dL) was applied instead of severe hypoglycemia (40 mg/dL), these results support the recommendation of a conventional glucose strategy over a strict glucose control strategy [50, 52].

In the comparison of outcomes among subgroups, both neurological outcome and mortality were highest in controlled diabetes patients and lowest in unrecognized diabetes patients, and the reason for this is unclear. The potential reasons for poor outcomes in controlled diabetes patients and good outcomes in unrecognized diabetes patients may be attributed to inadequate statistical power due to a small sample size and potential age bias in the study population. In univariate analysis, it was found that known diabetes, whether well controlled or not, had a positive effect on poor neurological outcome and mortality. Diabetes is a state of increased release of proinflammatory mediators and counterregulatory hormones, making

patients susceptible to infection, and it is highly exposed to microvascular or macrovascular injury through complex mechanisms, which can affect the results [17, 53]. The patient categories did not have an impact in multivariate analysis.

There are several limitations to our study. First, it is a single-center registry and retrospective study. Second, the number of patients with controlled diabetes was less than 10% of the total population, which may have limited our ability to make comparisons. Third, we did not include factors related to TTM in the logistic regression analysis, which may have influenced glucose hemodynamics. Forth, a substantial portion of data regarding oral medications or insulin usage before cardiac arrest was unavailable for patients diagnosed with diabetes, leading to an inadequate basis for analysis. Last, HbA1c has been used to predict recent glycemic status, but factors such as hemodialysis, recent blood loss, transfusion, or erythropoietin therapy that can alter the relationship between HbA1c and glycemia have not been investigated in this study. Additionally, classifying patients solely based on glycemic status is imperfect in the prediction of undiagnosed diabetes.

## Conclusion

Patients who survived after cardiac arrest showed differences in glucose hemodynamics and outcomes depending on their preexisting diabetes status and glycemic status. In particular, even a single occurrence of hypoglycemia during the acute period could have an impact on poor neurological outcomes. Although this categorization did not directly influence the neurological outcomes, we could suggest future studies are warranted to implement a tailored glucose control approach in each group.

## Supporting information

**S1 File. Patient information and data used in analysis.**
(PDF)

**S2 File. Glucose levels of patients by category used for Fig 2.**
(PDF)

**S1 Table. Univariate logistic regression analysis.**
(DOCX)

## Author Contributions

**Conceptualization:** Chun Song Youn, Kyu Nam Park, Sang Hoon Oh.

**Data curation:** Hyo Jin Bang, Sang Hyun Park.

**Formal analysis:** Hyo Jin Bang, Hyo Joon Kim, Sang Hyun Park.

**Investigation:** Hyo Jin Bang, Hyo Joon Kim, Soo Hyun Kim.

**Methodology:** Chun Song Youn.

**Supervision:** Chun Song Youn.

**Validation:** Kyu Nam Park.

**Writing – original draft:** Hyo Jin Bang.

**Writing – review & editing:** Chun Song Youn, Sang Hoon Oh, Soo Hyun Kim.

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
