## [Decision Letter · Decision Letter 0]

4 Dec 2023

PONE-D-23-24429Glucose control and outcomes in diabetic and nondiabetic patients treated with targeted temperature management after cardiac arrestPLOS ONE

Dear Dr. Youn,

Thank you for submitting your manuscript to PLOS ONE. After careful consideration, we feel that it has merit but does not fully meet PLOS ONE’s publication criteria as it currently stands. Therefore, we invite you to submit a revised version of the manuscript that addresses the points raised during the review process.

We look forward to receiving your revised manuscript.

Kind regards,

Amirmohammad Khalaji

Academic Editor

PLOS ONE

Journal Requirements:

Reviewers' comments:

Reviewer's Responses to Questions

**Comments to the Author**

1. Is the manuscript technically sound, and do the data support the conclusions?

Reviewer #1: Yes

Reviewer #2: Partly

2. Has the statistical analysis been performed appropriately and rigorously? 

Reviewer #1: Yes

Reviewer #2: No

3. Have the authors made all data underlying the findings in their manuscript fully available?

Reviewer #1: Yes

Reviewer #2: Yes

4. Is the manuscript presented in an intelligible fashion and written in standard English?

Reviewer #1: Yes

Reviewer #2: Yes

5. Review Comments to the Author

Reviewer #1: Dear authors, this is an interesting study regarding the impact of glycemic control in patients with pre existing DM and those without in neuroprotection after survivors of cardiac arrest.

- Introduction: could benefit from adding a hypothesis and to provide more scientific rationale for it. The following paragraph "In this study, we classified patients into two groups: those who had been diagnosed with diabetes and those who had not. We then further classified patients in these two groups into subgroups based on their preexisting levels of glucose control using glycated hemoglobin (HbA1c). Our final aim was to compare the glucose hemodynamics of these subgroups and, if differences exist, provide a basis for more ideal glucose control for each group. We used HbA1c to assess preexisting levels of glucose control [27–29]. A cutoff value ofHbA1c 6.0% was set to determine whether glucose levels were well controlled because we believed that different standards might be appropriate for acute-care settings such as emergency rooms." contains many elements that belong in the methods section. In the introduction it would be provides the hypothesis, study rationale and aims.

- Methods: Well written but I would add the information of the last paragraph of the introduction in the methods section. Please provide the reference for you cutoff for hypoglycemia. And please provide the rational for using HbA1c of 6 % instead of 6.5%. Statistical analysis: the multivariate analysis should also include known risk factors for poor outcome in cardiac arrest and not just with a p<0.05 in the univariate analysis as this approach is not recommended.(DOI 10.1513/AnnalsATS.201808-564PS)

- Results: The following sentence "HbA1c, admission glucose, mean and median values of initial glucose level were significantly higher in the groups of inadequately controlled diabetes and unrecognized diabetes."seems redundant since HbA1c was used to classify the different subgroups. It would also be interesting to know if the inulin doses were different in the groups regarding the use of oral DM medication. In the multivariable analysis I don't think the authors should add both the variable different subgroups and the variable HbA1c , since HbA1C was use to categorize the different subgroups

- discussion: well written. I have some comments : While I agree that stress induced hyperglycemia is a great component to justify hyperglycemia in post cardiac arrest and this should merits a paragraph I the discussion, this is true for all subgroups and I am not sure that it can be used to explain the differences in the difficult to achieve glycemic control the different groups. Another interesting point is that although all patient underwent TTM no data is given regarding body temperature and other TTM data as this may also have influenced glycemic metabolic. this is acknowledged by the authors in the limitations.

Reviewer #2: I have some critiques authors need to address.

• Objective is clear that authors categorized patients with glycemic condition and diabetes diagnosis. Then, authors evaluated association between outcomes and categorization as well as glycemic conditions. Conclusion is incomplete. Especially in the abstract section, it is vague what authors want to convey to the readers. Categorization was a primary objective. However, it did not predict outcomes. Authors should touch on the conclusions. Authors also should suggest or indicate treatment plan based on the discoveries of glycemic hemodynamics.

• Regarding authors discussion about worse outcomes in controlled diabetes patients and best outcomes in unrecognized diabetes patients, are the potential reasons inadequate power due to small sample size and age bias?

• How did Author define range glucose? It is not described in method section. Authors mention range glucose in discussion section with univariate analysis data only. Is there any reason why authors do not conduct multivariate analysis for this marker?

• CPC is not defined.

• Method section: “initial HbA1c of 6.0% or less” does this mean initial HbA1c less than 6.0%?

• Figure 2 does not have foot note. Are the data median +/- IQR?

6. PLOS authors have the option to publish the peer review history of their article (what does this mean?). If published, this will include your full peer review and any attached files.

Reviewer #1: No

Reviewer #2: No

---

## [Author Response · Author response to Decision Letter 0]

21 Dec 2023

Reviewer #1

───────────────────────────────────────────────────────────────

Dear authors, this is an interesting study regarding the impact of glycemic control in patients with pre existing DM and those without in neuroprotection after survivors of cardiac arrest.

1-1) Introduction: could benefit from adding a hypothesis and to provide more scientific rationale for it. The following paragraph "In this study, we classified patients into two groups: those who had been diagnosed with diabetes and those who had not. We then further classified patients in these two groups into subgroups based on their preexisting levels of glucose control using glycated hemoglobin (HbA1c). Our final aim was to compare the glucose hemodynamics of these subgroups and, if differences exist, provide a basis for more ideal glucose control for each group. We used HbA1c to assess preexisting levels of glucose control [27–29]. A cutoff value ofHbA1c 6.0% was set to determine whether glucose levels were well controlled because we believed that different standards might be appropriate for acute-care settings such as emergency rooms." contains many elements that belong in the methods section. In the introduction it would be provides the hypothesis, study rationale and aims.

2-1) Methods: Well written but I would add the information of the last paragraph of the introduction in the methods section. 

→ We appreciate the reviewer’s suggestion. We edited the introduction and the methods as you commented for the concision and precision of the article.

2-2) Please provide the reference for you cutoff for hypoglycemia. 

→ We thoroughly agree with the reviewer’s comment on the lack of the citation. We added the reference of the hypoglycemia criteria.

2-3) And please provide the rational for using HbA1c of 6 % instead of 6.5%. 

→ We appreciate the reviewer’s suggestion. It seems that the explanation as to why we set 6.0% as the standard was insufficient. According to the reviewer’s opinion for clarity, we tried to explain in more detail why we set 6.0% instead of 6.5%.

2-4) Statistical analysis: the multivariate analysis should also include known risk factors for poor outcome in cardiac arrest and not just with a p<0.05 in the univariate analysis as this approach is not recommended. (DOI 10.1513/AnnalsATS.201808-564PS)

→ We totally agree with the reviewer’s thoughtful comments about the multivariate analysis. As the reviewer suggested, we included a set of potential confounders derived from published studies. These confounders were witness status, bystander CPR, and the initial presence of a shockable rhythm. We performed multivariate logistic regression with all these variables. The results were edited and the method section was also supplemented. 

3-1) Results: The following sentence "HbA1c, admission glucose, mean and median values of initial glucose level were significantly higher in the groups of inadequately controlled diabetes and unrecognized diabetes."seems redundant since HbA1c was used to classify the different subgroups. 

→ We thoroughly agree with the reviewer’s point of view. We edited the sentences for brevity as the reviewer suggested and corrected the incorrectly mentioned word.

3-2) It would also be interesting to know if the inulin doses were different in the groups regarding the use of oral DM medication. 

→ HbA1c, Admission glucose, mean and median values of initial glucose level were significantly higher in the groups of inadequately controlled diabetes an We sincerely agree with you and thank you for your insightful comment. At first, we also wanted to analyze the oral medication or insulin that patients diagnosed with diabetes had previously received, but there was a lot of missing information, so we were unable to analyze the information in this study. Reflecting the thoughtful point, the following information was added to the discussion section.

3-3) In the multivariable analysis I don't think the authors should add both the variable different subgroups and the variable HbA1c, since HbA1C was use to categorize the different subgroups

→ We deeply appreciate the reviewer’s thoughtful comments. As suggested by the authors, HbA1C was excluded from the multivariate analysis as it was used to classify different subgroups. We re-conducted the statistical analysis, including known risk factors following the advice provided in points above (2-4). The results have been reflected in the Results section. 

4) Discussion: well written. I have some comments: While I agree that stress induced hyperglycemia is a great component to justify hyperglycemia in post cardiac arrest and this should merit a paragraph I the discussion, this is true for all subgroups and I am not sure that it can be used to explain the differences in the difficult to achieve glycemic control the different groups. Another interesting point is that although all patient underwent TTM no data is given regarding body temperature and other TTM data as this may also have influenced glycemic metabolic. this is acknowledged by the authors in the limitations.

→ We fully agree with you and thank you for the insightful comments. Our ultimate goal of categorization was to help achieve a better outcome by providing tailored glucose control based on patients’ glycemic status rather than a uniform treatment. We believe that further studies, including prospective studies or RCTs, will help better glucose control and improve outcomes if they secure the legitimacy of personalized treatment. 

Reviewer #2

───────────────────────────────────────────────────────────────

I have some critiques authors need to address.

1) Objective is clear that authors categorized patients with glycemic condition and diabetes diagnosis. Then, authors evaluated association between outcomes and categorization as well as glycemic conditions. Conclusion is incomplete. Especially in the abstract section, it is vague what authors want to convey to the readers. Categorization was a primary objective. However, it did not predict outcomes. Authors should touch on the conclusions. Authors also should suggest or indicate treatment plan based on the discoveries of glycemic hemodynamics.

→ We do appreciate the reviewer's suggestion on the vague conclusion. As per the reviewer's opinion, we strengthened the contents of the treatment on hyperglycemia in the discussion section and tried to mention the contents more clearly in the abstract and conclusion section.

2) Regarding authors discussion about worse outcomes in controlled diabetes patients and best outcomes in unrecognized diabetes patients, are the potential reasons inadequate power due to small sample size and age bias?

→ We fully agree with the thoughtful comments of the reviewer. Despite various factors influencing the neurological outcome, as the reviewer rightly pointed out, the small sample size and the increasing comorbidity with patient age might have had an impact on the outcome. We cautiously predicted that these factors could have influenced the outcome. We have incorporated the reviewer’s valuable input into the discussion section. 

3-1) How did Author define range glucose? It is not described in method section. 

→ As pointed out by the reviewer, the definition has been added to the method for better understanding. As per the reviewer's comment, we provided the definition of the range glucose in the revised manuscript.

3-2) Authors mention range glucose in discussion section with univariate analysis data only. Is there any reason why authors do not conduct multivariate analysis for this marker?

→ We deeply appreciate the reviewer’s thoughtful comments. In the univariate analysis considering individual factors, admission glucose, mean, median, and the range of glucose during the initial 48 hours remained statistically significant. However, when attempting a more comprehensive multivariate analysis, these variables lost statistical power, and thus, their results were not originally included in the manuscript before the revision. In response to the suggestions from Reviewer 1 and Reviewer 2 to incorporate established risk factors for adverse outcomes in cardiac arrest and variables with a p-value less than 0.05 in the univariate analysis, we conducted a multivariate analysis with these specified variables. The conclusive outcomes are presented in Table 4. To offer a thorough overview, we also included supplementary Table 1, delineating the findings from the univariate logistic regression analysis.

4) CPC is not defined.

→ We appreciate the reviewer’s comment on the lack of CPC definition. According to the reviewer's comment, we defined CPC in the method section. 

5) Method section: “initial HbA1c of 6.0% or less” does this mean initial HbA1c less than 6.0%?

→ We deeply appreciate the reviewer’s thoughtful comments. Generally, our center set the target temperature as 33°C. Ninety percent of the patients received TTM at 33°C, while the remaining received TTM at 36°C. As the reviewer suggested, we have added the target temperature column to the table included in the results. Additionally, we have mentioned the details in the revised manuscript.

6) Figure 2 does not have foot note. Are the data median +/- IQR?

→ We appreciate the reviewer’s comment on the lack of the footnote. We tried to show the glucose trend of each group, the data of Fig 2 are mean value with standard deviation. We added the footnote as figure legends in revised manuscript.

---

## [Decision Letter · Decision Letter 1]

29 Jan 2024

Glucose control and outcomes in diabetic and nondiabetic patients treated with targeted temperature management after cardiac arrest

PONE-D-23-24429R1

Dear Dr. Youn,

We’re pleased to inform you that your manuscript has been judged scientifically suitable for publication and will be formally accepted for publication once it meets all outstanding technical requirements.

Kind regards,

Amirmohammad Khalaji

Academic Editor

PLOS ONE

Additional Editor Comments (optional):

Reviewers' comments:

Reviewer's Responses to Questions

**Comments to the Author**

1. If the authors have adequately addressed your comments raised in a previous round of review and you feel that this manuscript is now acceptable for publication, you may indicate that here to bypass the “Comments to the Author” section, enter your conflict of interest statement in the “Confidential to Editor” section, and submit your "Accept" recommendation.

Reviewer #1: All comments have been addressed

Reviewer #2: All comments have been addressed

2. Is the manuscript technically sound, and do the data support the conclusions?

Reviewer #1: Yes

Reviewer #2: Yes

3. Has the statistical analysis been performed appropriately and rigorously? 

Reviewer #1: Yes

Reviewer #2: Yes

4. Have the authors made all data underlying the findings in their manuscript fully available?

Reviewer #1: Yes

Reviewer #2: (No Response)

5. Is the manuscript presented in an intelligible fashion and written in standard English?

Reviewer #1: Yes

Reviewer #2: Yes

6. Review Comments to the Author

Reviewer #1: Congratulations to the authors have answered all of the reviewers' queries. I endorse this manuscript for publication.

Reviewer #2: Authors should correct below point. Other than that, the manuscript is well rerevised and good shape for publication.

Line 213-215, Authors re-calculated multi-variate logistical regression. Yet the OR or CI numbers are not updated.

7. PLOS authors have the option to publish the peer review history of their article (what does this mean?). If published, this will include your full peer review and any attached files.

Reviewer #1: No

Reviewer #2: No

---

## [Editor Report · Acceptance letter]

31 Jan 2024

PONE-D-23-24429R1 

PLOS ONE

Dear Dr. Youn, 

I'm pleased to inform you that your manuscript has been deemed suitable for publication in PLOS ONE. Congratulations! Your manuscript is now being handed over to our production team.

Kind regards, 

on behalf of

Dr. Amirmohammad Khalaji 

Academic Editor

PLOS ONE